# Chronic Hyperkaliemia in Chronic Kidney Disease: An Old Concern with New Answers

**DOI:** 10.3390/ijms23126378

**Published:** 2022-06-07

**Authors:** Silvio Borrelli, Ida Matarazzo, Eugenio Lembo, Laura Peccarino, Claudia Annoiato, Maria Rosaria Scognamiglio, Andrea Foderini, Chiara Ruotolo, Aldo Franculli, Federica Capozzi, Pavlo Yavorskiy, Fatme Merheb, Michele Provenzano, Gaetano La Manna, Luca De Nicola, Roberto Minutolo, Carlo Garofalo

**Affiliations:** 1Silvio Borrelli Unit, Nephrology Department of Advanced Medical and Surgical Sciences, University of Campania “Luigi Vanvitelli” Piazza Miraglia, 80138 Naples, Italy; ida.matarazzo@gmail.com (I.M.); eugeniolembo1887@gmail.com (E.L.); peccarinolaura@hotmail.it (L.P.); claudia.annoiato@gmail.com (C.A.); mariarosaria.scognamiglio3@gmail.com (M.R.S.); andreafode@gmail.com (A.F.); chiara.ruotolo@yahoo.it (C.R.); francullialdo95@gmail.com (A.F.); federica.capozzi@gmail.com (F.C.); paolo.yavorskiy@gmail.com (P.Y.); fatme.mrb.177@gmail.com (F.M.); luca.denicola@unicampania.it (L.D.N.); roberto.minutolo@gmail.com (R.M.); carlo.garofalo@unicampania.it (C.G.); 2Nephrology, Dialysis and Transplant Unit of the University of Bologna “Alma Mater Studiorum”, 40126 Bologna, Italy; michiprov@hotmail.it (M.P.); gaetano.lamanna@unibo.it (G.L.M.)

**Keywords:** potassium, end-stage kidney disease, chronic kidney disease

## Abstract

Increasing potassium intake ameliorates blood pressure (BP) and cardiovascular (CV) prognoses in the general population; therefore the World Health Organization recommends a high-potassium diet (90–120 mEq/day). Hyperkalaemia is a rare condition in healthy individuals due to the ability of the kidneys to effectively excrete dietary potassium load in urine, while an increase in serum K^+^ is prevalent in patients with chronic kidney disease (CKD). Hyperkalaemia prevalence increases in more advanced CKD stages, and is associated with a poor prognosis. This scenario generates controversy on the correct nutritional approach to hyperkalaemia in CKD patients, considering the unproven link between potassium intake and serum K^+^ levels. Another concern is that drug-induced hyperkalaemia leads to the down-titration or withdrawal of renin-angiotensin system inhibitors (RASI) and mineralocorticoids receptors antagonists (MRA) in patients with CKD, depriving these patients of central therapeutic interventions aimed at delaying CKD progression and decreasing CV mortality. The new K^+^-binder drugs (Patiromer and Sodium-Zirconium Cyclosilicate) have proven to be adequate and safe therapeutic options to control serum K^+^ in CKD patients, enabling RASI and MRA therapy, and possibly, a more liberal intake of fruit and vegetables.

## 1. Introduction

Potassium plays a fundamental role in the correct functioning of human cells, regulating the resting membrane potential of excitable cells and blood pressure. The World Health Organization recommends a potassium intake of 90–120 mEq/day for individuals with normal renal function to prevent high blood pressure, osteoporosis, and kidney stones [1]. A recent meta-analysis of six observational studies including 10,709 patients showed that each one-gram increase in potassium intake is associated with an 18% reduction in cardiovascular (CV) risk [2]. Furthermore, in the general population, the use of salt substitute (25% KCl, 75% NaCl) vs. regular salt (100% NaCl) is associated with improved BP and reduced incidence of CV outcomes with no increased significant increase in serum K^+^ [3]. However, although higher potassium intake decreases BP levels and is associated with lower mortality, increased serum potassium is associated with a greater mortality and higher risk of end-stage kidney disease (ESKD) [4]. Hyperkalaemia is a rare condition in healthy individuals due to the ability of the kidneys to effectively excrete potassium in urine (potassium adaptation), while it becomes common in patients with chronic kidney disease (CKD), mainly due to the reduction of renal function, metabolic acidosis, and the wide use of renin-angiotensin system inhibitors (RASI) [4].

This review will consider new potential answers to hyperkalaemia, while also considering novel insights into the relationship between potassium intake and blood pressure (BP), and the unproven link between potassium intake and hyperkalaemia.

## 2. Potassium Homeostasis

On average, the total body potassium content of an adult human is 50–75 mEq/kg body weight (BW); over 98% of the entire body’s potassium is contained in the cells (muscles, liver, red blood cells), and only a tiny fraction is extracellular. In normal conditions, serum K^+^ is constantly maintained within normal ranges (4.0–4.9 mEq/L) independent of potassium intake, through multiple mechanisms. The redistribution of potassium between the intra- and extracellular compartments provides the first line of defence against variations in extracellular K^+^ concentration (internal potassium balance). In subjects with normal renal function, the maintenance of external potassium balance depends primarily on excretion by the kidneys, because the amount excreted in the faeces is only 5–10% of the potassium intake (Figure 1) [1]. 

### 2.1. Internal Potassium Balance

The internalization of potassium into the cells occurs through an Na^+^/K^+^ ATPase-dependent pump that enables potassium uptake into the cells. Na^+^/K^+^ ATPase-dependent pump activation is chiefly regulated by insulin and catecholamines.

Briefly, an increase in serum K^+^ (e.g., after a meal) induces the secretion of insulin, which is the primary regulator hormone that promotes a shift of potassium into the liver and muscle cells by increasing Na^+^/K^+^ ATPase-dependent pump activity. Furthermore, increased serum K^+^ causes the release of catecholamines, which shift potassium into the cells via β-2 adrenergic receptors. These hormones act permissively, since increased serum K^+^ levels do not induce their secretion. Additionally, aldosterone promotes potassium uptake into the cells, though it mainly acts on renal potassium excretion. 

Again, acid–base balance affects serum potassium by exchanging hydrogen cations for potassium across the cell membrane: acidaemia induces the shift of potassium out of the cell, whereas the reverse occurs during alkalemia. Although acidaemia tends to increase serum K^+^, this effect is more pronounced in organic acidosis than in metabolic acidosis. Indeed, metabolic acidosis due to inorganic acids causes an intracellular reduction of Na^+^, owing to the reduced activity of Na^+^/H^+^ exchange and Na^+^/Bicarbonate cotransport; reduced intracellular Na^+^ induces a decrease of the Na^+^/K^+^ ATPase pump, causing, in turn, the outflow of K^+^ from cells. Instead, in metabolic acidosis due to organic acids, these anions and H^+^ enter the cells through monocarboxylate transporters. The increase of intracellular H^+^ concentration stimulates the activity of Na^+^/H^+^ exchange, which accumulates intracellular Na^+^, maintaining Na^+^/K^+^ ATPase pump activity; thus, serum K^+^ levels change minimally.

Additional factors increasing extracellular potassium concentration are high plasma osmolarity, prolonged physical exercise, and cellular lysis (Figure 1) [5].

### 2.2. External Potassium Balance

The kidneys excrete 90–95% of potassium intake, representing the primary regulator of potassium homeostasis. In normal conditions, extra-renal potassium excretion is principally mediated by gastrointestinal excretion, accounting for 5–10% (Figure 1). The potassium excretion by the kidney is the sum of filtration and tubular potassium handling (tubular reabsorption and secretion). The amount of filtered potassium is merely the product of serum potassium and glomerular filtrate rate (e.g., 4.2 mEq/L × 180 L/day = 720 mEq/day). Under normal conditions, about 65% of the filtered potassium is reabsorbed by the proximal tubule, while 25–30% is reabsorbed by the thick ascending loop, which reabsorbs via the Na^+^/K^+^/2Cl^−^-cotransporter. This fraction of potassium absorption in the proximal tubules is relatively constant. Day-to-day potassium regulation depends on the secretion in the principal cells of late distal tubules and cortical collecting tubules (aldosterone-sensitive nephron tracts). In some circumstances, distal potassium secretion can be counterbalanced by potassium reabsorption mediated by the intercalated cells in downstream tubules (the cortical and outer medullary collecting tubules). The potassium reabsorption is mediated by an active H^+^-K^+^-ATPase pump in the luminal membrane, which results in K^+^ reabsorption and H^+^ secretion [5]. 

## 3. Regulation of Urinary Potassium Excretion

Renal potassium excretion is regulated by a negative-feedback system that regulates urinary potassium excretion in response to small serum K^+^ change. Increased serum K^+^ reduces thiazide-sensitive Na^+^/Cl^−^ cotransporters (NCC) activity in the proximal portion of the distal convoluted tubule (DCT), increasing Na^+^ delivery in the distal aldosterone-sensitive portions of the cortical collecting duct (CCD). Furthermore, increased serum K^+^ stimulates aldosterone secretion by the adrenal gland, which induces Na^+^ reabsorption by the amiloride-sensitive epithelial sodium channel (ENaC). Therefore, aldosterone-mediated tubular potassium excretion is mainly driven by the lumen-negative potential generated by Na^+^ absorption. Additionally, Na^+^/K^+^-ATPase activity is enhanced by aldosterone. The renal K^+^ secretion occurs by two types of apical K^+^ channels: renal outer medullary potassium (ROMK) and Maxi-K^+^ channels (or Big K, BK). Under normal conditions, ROMK is the main regulating pathway of K secretion, while BK is activated by increased urinary flow.

Notably, hyperkalaemia stimulates aldosterone, increasing urinary potassium excretion without salt retention. In contrast, under conditions of volume depletion, aldosterone induces sodium absorption without change in urinary potassium excretion. This phenomenon is called the “aldosterone paradox”. Indeed, volume depletion causes the reduction of sodium delivery and tubular flux, and the increase of Angiotensin II levels. Angiotensin II blocks ROMK channels on the distal tubule and promotes aldosterone activity on the apical proton pumps (H^+^-ATPase and H^+^/K^+^-ATPases) and the pendrin (Cl^−^/HCO3^–^ exchanger) on intercalated cells, which increases potassium reabsorption [5]. Urinary potassium excretion is regulated by at least two other systems acting independently from serum K^+^ levels: a feed-forward system that induces kaliuresis in response to a meal in the absence of serum K^+^ elevation, which is triggered by the activation of gut potassium receptors, and a circadian system that works independently from potassium intake and serum K^+^ changes regulated by the suprachiasmatic nucleus in the brain [6]. The circadian rhythm of renal potassium handling works by lowering excretion at night and increasing it during the day, which seems to coincide with the timing of the consumption of K^+^-containing foods. This excretory potassium pattern reflects a circadian rhythm in plasma aldosterone levels and gene expression the of proteins that regulate potassium excretion [6]. The net result of potassium homeostasis is, therefore, that serum K^+^ is tightly maintained in narrow limits by urinary K^+^ excretion in response to potassium intake, which is usually higher than the K^+^ concentration in extracellular fluid (Figure 1) [5].

## 4. Molecular Mechanisms Explaining the Relationship between Potassium Intake and Blood Pressure

Novel experimental insights on the relationship between potassium intake and BP indicate that thiazide-sensitive NCC plays a pivotal role in regulating potassium excretion and increasing BP levels. Low potassium intake induces NCC activation, which induces Na^+^ reabsorption, thereby increasing BP [7]. Indeed, low extracellular K^+^ induces the outflow to the cells via ROMK (Kir 4.1/Kir 5.1) expressed on the basolateral membrane of DCT [8]. K^+^ outflow induces Cl^−^ efflux by ClC-Kb [9]. The intracellular Cl^−^ reduction causes the activation of WNK-Kinase 4, which activates phosphorylation NCC by STE20-related proline/alanine-rich kinase (SPAK) and oxidative-stress-responsive kinase 1 (OSR1) [10]. NCC phosphorylation increases Na^+^ and Cl^−^ reabsorption, reducing sodium delivery in the more distal segment of tubules. The consequent reduction in lumen electronegativity reduces potassium excretion at the level of CCD. NCC activity in DCT is enhanced when low potassium intake is associated with high sodium intake [7]. 

High K^+^ load reduces NCC activity (thiazide-like effect) by NCC dephosphorylation, which is mediated by a different molecular pathway (not Cl^−^ dependent). The increased sodium delivery to the downstream segments (CCD) promotes potassium excretion. A possible molecular mechanism is that the increase of extracellular potassium might induce a Ca^2+^ influx that activates the calmodulin-calcineurin pathway; the activated calcineurin induces NCC activity, resulting in a reduction of sodium reabsorption of DCT; however, this model has not been demonstrated experimentally [7]. 

In 5/6 nephrectomised rats, model simulation of potassium load induces down-regulation of NCC that may contribute to the increase of tubular K secretion, together with an increase in Na^+^/K^+^ ATPase dependent activity and expression in ROMK and BK channels [11].

Further proof of the importance of NCC expression is the recent finding that female rats showed an increased expression in NCC compared to males, which seems to be related to oestrogens and prolactin. This different NCC activity between males and females may provide a rationale for observing lower serum K^+^ in response to an oral potassium load in females than in males. The enhanced NCC expression also suggests a role in the physiological condition of salt retention during pregnancy and lactation [12].

## 5. Hyperkalaemia in General Population and in Chronic Kidney Disease

The risk of hyperkalaemia is negligible when kidney function is normal. The Chronic Kidney Disease Prognosis Consortium study, including 27 cohort studies (N = 1,217,986), showed that the prevalence of hyperkalaemia (serum K > 5.5 mmol) was 0.49% in the cohorts of the general population and those at higher CV risk [17 cohorts; N = 1,175,816 individuals], while it was more prevalent (4.2%) in the cohorts with CKD (10 cohorts, N = 42,170 patients). In the cohorts of patients with no overt CKD at baseline, the risk of hyperkalaemia exponentially increased for lower GFR [4]. The prevalence of chronic hyperkalaemia estimated in cohorts of CKD patients ranged from 5.6% in patients with a mean eGFR of 62 mL/min/1.73 m^2^ to over 50% in the more advanced CKD stages [13,14,15,16,17,18,19,20,21,22,23,24,25,26,27,28]. 

The increased prevalence of persistent hyperkalaemia in CKD patients is due to the inability of kidneys to excrete dietary K load. The capability to increase potassium excretion as GFR declines could be maintained until the advanced CKD stages unless other factors intervene to modify renal adaptation. Indeed, multiple factors can induce hyperkalaemia in CKD patients, including reduced aldosterone effect (use of RASI or MRA, renal tubular acidosis, hyporeninaemic hypoaldosteronism), reduced potassium cell uptake (insulin deficiency, metabolic acidosis, hyperglycaemia, uremic toxins), or reduced delivery of sodium and water in the distal tubules (heart failure, hypovolemia) [1,5]. 

Notably, the Consortium study participants in the cohorts of the general population and at higher risk followed up for an average of 6.9 years, and showed an increased risk of ESKD and CV mortality starting from serum K^+^ only mildly altered [4]. In our pooled analysis of four cohorts in 2443 patients with non-dialytic CKD (mean eGFR: 35 ± 17 mL/min/1.73 m^2^) regularly followed in forty-three renal clinics, we found that a mild hyperkalaemia (serum K^+^ ≥ 5.0 mmol/L) was present in 39% (95% CI: 37–41%) of patients, and this prevalence did not change after one year (37%, 95% CI: 37–41%; *p* = 0.266). In a median of 3.6 years of follow-up, we found an increased cumulative incidence of dialysis initiation associated with persistent hyperkalaemia in both visits (sub-hazard ratio, sHR: 1.27, 95% C.I.: 1.02–1.58), and new-onset hyperkalaemia at the second visit (sHR: 1.35, 95% C.I.: 1.05–1.72) when compared to the patients with absent hyperkalaemia in both visits [24]. We also estimated that maintaining normokalaemia is associated with a consistent cost saving due to delayed dialysis initiation (on average by 2.3 years) and improved survival (on average by 1.7 years) [29].

The clinical significance of exposure to chronic hyperkalaemia is still debated. Indeed, the plausibility of a causal relationship between increased serum K^+^ levels and adverse outcomes has not yet been demonstrated. Although hyperkalaemia may increase the risk of electrocardiography (ECG) abnormalities (peaking T waves, QRS prolongation and PR shortening), whether the correction of chronic hyperkalaemia can reduce mortality and delay the onset of dialysis remains unknown. In this regard, the recent development of a deep-learning model may better classify hyperkalaemia from the ECG abnormalities in patients with chronic kidney disease by recognizing patients at higher risk of arrhythmias. The application of a deep-learning model using ECG may enable screening for hyperkalaemia [30].

Nonetheless, the risk of hyperkalaemia in patients with CKD leads physicians to down-titrate or withdraw RASI [31,32,33,34], thus depriving CKD patients of the well-recognized protective effects of these drugs [35,36]. Indeed, several studies reported that RASI was underutilized in CKD patients, and that their rate of prescription was even lower for those in severe CKD stages. Notably, CKD patients who discontinued RASI experienced higher CV events in CKD patients [31,32,33,34].

## 6. The Dilemma of Nutritional Approach in Chronic Kidney Disease Patients

The benefits of higher potassium intake on hypertension and CV risk in the general population conflict with the increased risk of CV death and ESKD associated with increased serum K^+^ [1,2,3,4,5]. In the general population, potassium supplementation is associated with no increase in serum K^+^: a pooled analysis of 11 studies showed an increase of 0.14 (95% C.I.: 0.09–0.19) mmol/L in response to a mean potassium supplementation of 45.8 mmol/die (22–140 mmol/day) [37]. In contrast, recent findings of a 2-weeks run-in phase of a randomized trial in 191 patients with stage 3b-4 CKD have showed a mean increase in serum K^+^ by 0.4 mmol/L in the patients treated with 40 mmol/day of potassium chloride supplementation, with no change in BP and eGFR [38]. 

Therefore, an overall strategy for lowering serum K^+^ is to recommend reducing the intake of potassium-rich foods in CKD patients, though a potassium-restricted diet has not been yet validated [39,40]. Moreover, a relationship between potassium intake and serum K^+^ has never been demonstrated. Indeed, the studies evaluating potassium intake by urinary potassium excretion or food frequency questionnaire have shown a weak correlation between potassium intake and serum K^+^ concentrations in patients with non-dialytic CKD (r = 0.05) [41] and ESKD [25,42,43,44]. This poor correlation between serum and potassium intake is likely due to an increase in gastrointestinal excretion and intracellular redistribution. The faecal potassium excretion between healthy subjects and ESKD patients showed that potassium excretion in stool was three times higher in ESKD patients than in controls. After intraluminal BaCl_2_ administration, which blocks K^+^ channels (ROMK) in gastrointestinal cells, the increase in intestinal potassium excretion in ESKD is due to increased potassium secretion into the gut [45].

Moreover, studies evaluating the association between urinary potassium excretion and mortality provided contrasting findings on the possible role of higher potassium intake on mortality in CKD patients [46,47]. 

Again, a plant-dominant diet, which is a dietary source of potassium, has been associated with improved renal outcomes in the general population [48,49]; and, furthermore, potassium-rich healthy dietary patterns are associated with a reduced risk of death and ESKD in CKD patients [50,51,52]. Finally, in contrast with general belief, more significant sources of dietary potassium are animal-derived processed foods and food additives (e.g., preservatives), rather than vegetables and fruits [53,54].

In Figure 2 we provide a possible rationale for a more liberal use of a plant-dominant diet to improve hyperkalaemia. Restricted high-fibre food intake may alter the intestinal microbiota in patients with CKD, with a consequent increase in intestinal permeability and systemic inflammation. A high-fibre plant diet facilitates the proliferation of saccharolytic bacteria at the expense of proteolytic bacteria, consequently reducing uremic toxins and inflammation [55]. Diets rich in fruit and vegetables prevent stypsis and constipation, which may exacerbate hyperkalaemia, depending on gastrointestinal excretion and faecal volume [53,54]. A plant-dominant diet containing high alkali levels may improve metabolic acidosis [56].

On the other hand, the status of knowledge does not allow the liberalisation of dietary potassium intake in CKD patients, especially in anuric ESKD patients [57]. Nonetheless, fruit and vegetables extremely rich in potassium (e.g., sweet potatoes, pumpkin, soy, pineapples, bananas) and oxalates (e.g., star fruit, spinach, broccoli) should be avoided in ESKD patients. In these patients, a careful evaluation of the type of plant food is advisable to evaluate the potassium–fibre ratio. The nutritional approach should provide a patient-tailored diet that ensures adequate protein-caloric intake, high fibre intake, and reduced net fixed-acid production [53].

## 7. Role of K-Binders on Hyperkalaemia in Chronic Kidney Disease Patients

Beyond the restricted-potassium diet, the nephrological management of hyperkalaemia in CKD includes potassium binder agents, a counter-ion exchanging cations with K^+^, thus increasing the faecal excretion of potassium in foods [58]. Until a few years ago, sodium polystyrene sulfonate (SPS) was the only K^+^ binder available, a resin exchanging non-selectively Na^+^ with K^+^. However, patients scarcely tolerated SPS and they were poorly prescribed by Nephrologists [42,59,60]. Our longitudinal analysis evaluating the management of hyperkalaemia in 592 patients with CKD (mean eGFR: 40 mL/min/1.73 m^2^), regularly followed in six nephrology clinics, showed that serum K control did not modify over 12 months, persisting through roughly a third of the cohort. We found that potassium intake was low (about 50 mmol during 12 months of follow-up), whereas traditional K binders were poorly used (<8% after 12 mo.) [42]. The poor use of SPS was likely due to the higher risk of severe gastrointestinal events associated with these drugs, which has been recently reported in two large surveys [59,60]. In a Canadian retrospective matched-cohort study, 20,000 SPS users showed a greater incidence of gastrointestinal events [59]. Similarly, in a Swedish observational study carried out in almost 20,000 patients with CKD stages 4–5, the initiation of SPS (N = 3690) was associated with severe gastrointestinal adverse events (e.g., ischemic colitis, perforations) [60]. This risk of gastrointestinal events is enhanced by the concomitant use of sorbitol to contrast the constipation associated with SPS use [60].

The recent advent of novel K^+^ binders could improve hyperkalaemia management with positive consequences on RASI therapy and the nutrition of CKD patients. Patiromer sorbitex calcium and sodium zirconium cyclosilicate (SZC) have been recently approved for hyperkalaemia treatment [58].

### 7.1. Patiromer

Patiromer is a non-absorbable synthetic polymer made of smooth spherical 100 μm beads, which exchanges calcium ions for potassium, magnesium, and sodium (non-selective) in the colon. Being sodium-free, patiromer is also suitable for patients with fluid overload. The clinical trials PEARL-HF, OPAL-HK, and AMETHYST-DN, demonstrated the safety and efficacy of patiromer in the treatment of hyperkalaemia [61,62,63]. These trials showed that in patients with chronic hyperkalaemia, patiromer, 8.4 to 16.8 g once daily, reduced serum K^+^ in patients with CKD, diabetes mellitus type 2, and heart failure. The AMETHYST-DN trial showed that up to 95% of treated patients had normokalaemia over 52 weeks of follow-up, and 100% continued RASI throughout the trial [64,65]. Post-hoc analysis in a sub-cohort of 62 elderly patients (>75 years) with stage 3 CKD (mean eGFR: 42 mL/min/1.73 m^2^) associated with diabetes, showed that the use of patiromer was associated with hyperkalaemia control (serum K^+^ < 5.5 mmol/L) in 100% of patients after 4 weeks and lasted for 52 weeks. No serious adverse reactions occurred [64].

More recently, the AMBER trial showed that in two hundred ninety-five CKD patients affected by resistant hypertension, patiromer allowed for the maintenance of spironolactone (treatment of choice in this condition) in 86% (vs. 66% placebo) of patients over 12 months [65].

Finally, the findings of the DIAMOND trial have been recently reported. Eight hundred sixty-eight patients with heart failure with reduced ejection fraction (HFrEF) were enrolled in this trial. In the run-in phase, the treatment with RASI, mineralocorticoid-receptor antagonist (MRA), or angiotensin receptor–neprilysin inhibitor (ARNI) was initiated or optimized, combined with patiromer. After the run-in phase, patients were randomized to continue or stop patiromer. After 13 weeks, the group for which patiromer was allowed to optimize RASI therapy showed a lower incidence of hyperkalaemia [66]. The results of the DIAMOND trial suggest a possible combined use of new K binders with Finerenone, a novel nonsteroidal MRA that has recently shown a significant benefit on renal and CV outcomes, but with an increased risk of hyperkalaemia [67,68].

The recommended dose of patiromer is 8.4 g once daily, titrable up to 16.8 g. The most common adverse reactions include gastrointestinal events (e.g., nausea/ vomiting, constipation, and flatulence) and electrolyte disorders (e.g., hypercalcemia and hypomagnesemia).

### 7.2. Sodium Zirconium Cyclosilicate (SZC)

SZC is a non-absorbed, non-polymer inorganic cation-exchange crystalline compound, highly selective for monovalent cations, specifically potassium and ammonium ions. SZC starts to bind potassium in the small intestines; thus, a significant serum K+ reduction is evident within 1 h of SZC administration, and could be preferred in acute hyperkalaemia. SZC is calcium-free while containing 400 mg of sodium for each 5 g [58].

The clinical trials have documented a dose-dependent effect of SZC on reducing serum K levels. In the phase 3 HARMONIZE study, 84% of patients normalized serum K levels within 24 h, and 98% within 48 h, with the administration of 30 g SZC [69]. After 28 days, serum K + levels were significantly lower in patients randomized to the three-dose SZC groups (5, 10, or 15 g) than placebo [70]. The HARMONIZE-Extension study also showed that serum K+ was maintained below 5.1 mmol/L in 88% of treated patients over 11 months, regardless of the presence at baseline of CKD, heart failure, diabetes, or use of RASI inhibitors [71]. 

In the DIALIZE study, HD patients were randomized to treatment with SZC 5 g/day or placebo on non-dialysis days, with a 5 g dose titration up to 15 g to maintain normal serum K^+^ levels for four weeks. At the end of the dose adjustment period, 37%, 43%, and 19% of patients took SZC at doses of 5 g, 10 g, and 15 g, respectively. The percentage of patients who maintained a pre-dialysis serum K + level between 4.0 and 5.0 mmol/L was 41.2% in the SZC group and 1.0% in the placebo group (*p* < 0.001) [72]. 

The recommended dose of SZC is 10 g × 3/day for 48 h and 5 g/day during the maintenance phase, with a weekly 5 g dose titration, ranging from a minimum dose of 5 g every other day to a maximum dose of 15 g/day. The recommended dose of SZC is 5 g to titrate to 15 g on non-dialysis days. The main adverse reactions are mild or moderate oedema at a high dose (30 g of SZC corresponds to 2400 mg in sodium) and gastrointestinal disorders (flatulence, constipation, and nausea) [58].

### 7.3. Strategies to Remove Potassium in Dialysis Patients

The whole potassium content in ESKD patients is normal or reduced, suggesting that an impaired cellular potassium uptake due to reduced muscle mass, metabolic acidosis, altered catecholamine signalling, and insulin resistance can increase serum K^+^ levels [73]. Although potassium excretion by the gastrointestinal system increases threefold in ESKD patients compared to healthy individuals [45,74], the imbalance between potassium intake and removal by dialysis is the leading cause of hyperkalaemia in dialysis patients. Accordingly, potassium intake recommended in anuric ESKD patients is 50–75 mmol/day (corresponding to a daily potassium intake of 2–3 g) [1].

Potassium has a small molecular weight (39.1 units); therefore, dialysis mainly removes potassium from the bloodstream by diffusion. Potassium removal is typically 30–50 mmol/day regardless of dialysis modality, though IHD is preferable for rapid correction of acute hyperkalaemia. 

In peritoneal dialysis (PD), dialysate bags typically are free of potassium; thus, the potassium gradient (difference between serum and dialysate concentration) corresponds to the serum K^+^ concentration. Accordingly, the removal of potassium by the convective mechanism is negligible because serum K^+^ concentration is low. PD is a “continuous” treatment that may improve hyperkalaemia management compared to intermittent dialysis (incremental PD or haemodialysis). 

Intermittent haemodialysis (IHD) efficiently removes potassium from blood in the first hours of treatment, lowering serum K^+^ by approximately 1 mEq/L per hour in the first two hours; then, serum K^+^ correction decreases gradually [75]. The rate of the potassium correction depends on the potassium concentration in the dialysate (potassium gradient). Considering that a rapid drop in serum K^+^ may predispose to ECG changes and arrhythmias, the optimum dialysate potassium prescription is still challenged [76,77]. According to the current recommendations, the dialysate potassium content must usually be 2–3 mEq/L for each IHD session, targeting a pre-dialysis serum K^+^ < 5.5 mmol/L and a post-dialysis serum K^+^ > 3.0 mmol/L, avoiding a reduction in serum K^+^ > 1 meq/L/hour [78]. Again, potassium fluctuations during HD have been correlated with a lengthening in QT interval [79] and higher QT dispersion [80], increasing the risk of arrhythmias. Alternative dialytic strategies, e.g., acetate-free biofiltration with K-profile, have been proposed to maintain a constant potassium gradient during haemodialysis [81], though a long-term benefit has not yet been demonstrated.

The rebound of serum K^+^ occurs within the following six hours after IHD, attenuating by approximately 70% the serum K^+^ decline achieved in dialysis [82]. Plasma tonicity is a critical factor of the post-dialysis serum K^+^ rebound, as reported by De Nicola et al., that showed a more significant serum K^+^ increase throughout the interdialytic period in those patients who underwent high sodium dialysate concentration (Na^+^: 143 mmol/L) than patients who were treated with low dialysate concentration (Na^+^: 138 mmol/L), despite comparable dialytic adequacy (e.g., KtV), intradialytic potassium removal, and inter-dialytic protein intake [83]. Higher potassium rebound was likely induced by intracellular fluid volume/extracellular volume redistribution of water and potassium, as previously reported in patients with non-dialytic CKD [84].

## 8. Conclusions

In hypertensive individuals, potassium supplementation improves CV prognosis by reducing BP levels, though the increase in serum K (5.0 mmol) is associated with greater mortality in general population cohorts. Hyperkalaemia is a rare condition when kidney function is normal, while it becomes more frequent as GFR declines. Possible consequences of chronic hyperkalaemia are the down-titration/suspension of RASI or the reduction of K+-rich food (fruits and vegetables), depriving CKD patients of essential tools to decrease their greater CV risk. The new K+-binder drugs (Patiromer, Sodium-Zirconium Cyclosilicate) provide adequate and safe therapeutic options to reduce serum potassium in CKD patients, allowing them to maintain RASI treatment and potentially an adequate intake of fruit and vegetables in CKD patients. 

## Figures and Tables

**Figure 1 ijms-23-06378-f001:**
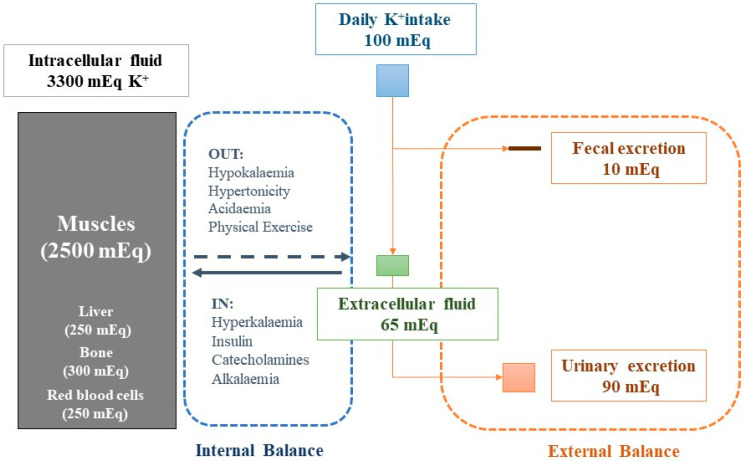
Potassium intake recommended by guidelines (blue box) and body potassium distribution between intracellular (grey box) and extracellular fluid (green box). Graph illustrates internal (dashed blue box) and external (dashed orange box) potassium homeostasis. Colored boxes’ sizes are proportionate to potassium content.

**Figure 2 ijms-23-06378-f002:**
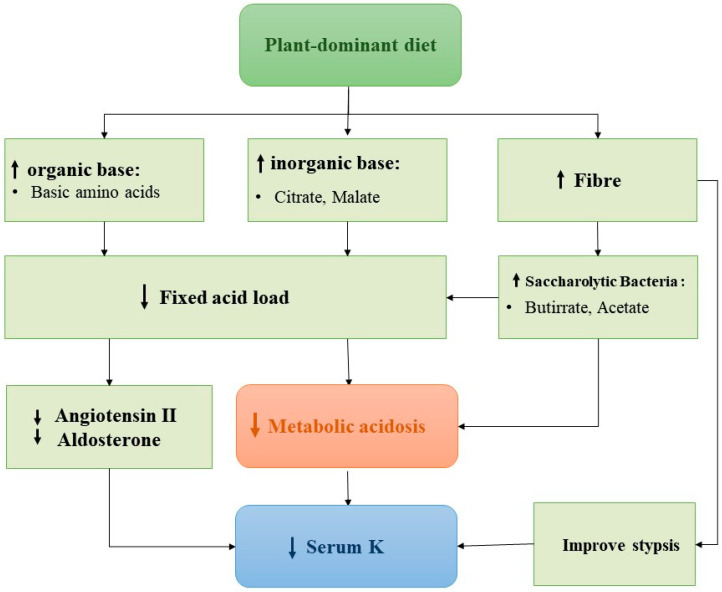
Possible positive effects on serum K^+^ related to plant-dominant diet.

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
