# Peer review of "Chronic Hyperkaliemia in Chronic Kidney Disease: An Old Concern with New Answers"

_ijms, 2022, doi:10.3390/ijms23126378_

Round 1
Reviewer 1 Report
The manuscript titled “Chronic hyperkalaemia in chronic kidney disease: an old question with new answers” In general, the paper do not provide any new information about molecular diagnostics for hyperkalaemia; The major concern is that this report is only a clinical review, it indeed provides less information about the advance of molecular information of hyperkalaemia in patients with chronic kidney disease.
Author Response
We are grateful to the reviewer for his comment.
According to his suggestion, we have profoundly modified the manuscript, including a new paragraph (n.4) entitled: "Molecular mechanisms explaining the relationship between potassium intake and blood pressure".
We have enriched the manuscript with details regarding molecular mechanisms and reduced the clinical findings. The reviewer can find all text modifications in the PDF with track changes.
Thank you,
Dr Silvio Borrelli
Nephrology Unit. University of Campania "Luigi Vanvitelli", Naples (Italy)
Reviewer 2 Report
This is a well-structured work that explains in detail the role of potassium in chronic kidney disease and the new therapeutic approaches aimed at reducing K+ levels in the blood.
I think the information is complete, I only have a curiosity that has not been resolved after reading the manuscript and it is the role of sweat in potassium excretion. Although I understand that the amount may not be relevant, it is possible that certain lifestyles (sport vs. sedentary lifestyle) can also influence the amount of K+ in the blood. Is there anything known about this topic?
Regarding formatting issues, in point 3 there are several paragraphs that start with a different spacing than the rest.
In figure 1 I would remove the mention of K+ in intracellular fluid and daily intake since the rest of the values do not mention it. If this were done, it should be made clear in the figure caption that all values refer to potassium. Another option is to add to all the values the K+
Author Response
We are grateful to the reviewer for the positive comments.
As suggested in the first comment, we have added a specific comment on the loss of potassium by sweating (lines 89-90 of the updated version).
We have removed extra spaces and formatting errors. Thank you.
Regarding figure 1, we have enriched the text of values and modified the figure caption.
Round 2
Reviewer 1 Report
Dear authors: Thank for your revision!
In the manuscript, the authors only conclude hyperkalaemia may increase the risk of electrocardiography (ECG) abnormalities, whether the correction of chronic hyper- kalaemia can reduce mortality and delay the onset of dialysis remains unknown. ECGs are commonly used as point-of-care (POC) tests to measure the electrical activity of the heart. Certain electrical changes in the ECG have been associated with dyskalaemia, which can be confirmed by laboratory examination. However, prompt recognition of dyskalaemia-associated ECG changes prior to laboratory results is still fraught with great challenge in emergent situations. It may be helpful to provide some information for the readers of IJMS about this topic in your revised manuscript.
Sincerely, Po-Jen Hsiao, MD
Division of Nephrology, Department of Internal Medicine, Taoyuan Armed Forces General Hospital, Taoyuan, Taiwan
Email: a2005a660820@yahoo.com.tw or doc10510@aftygh.gov.tw
Author Response
We are grateful to the reviewer for the suggestions, which allow improving our manuscript. In the new reviewed paper, we have added a new paragraph about the use of deep learning to recognize patients at higher risk of arrhythmogenic hyperkalemia (lines 206-15) with new reference (DOI:10.1001/jamacardio.2019.0640.).
Thank you again.